

# Greenland's Topography Triggers Cyclogenesis: Synergy between Lee Cyclogenesis and Jet Streak

**Cheng You[1, 2]**

[1]Alfred Wegener Institute, Helmholtz Centre for Polar and Marine Research, Potsdam, Germany

[2] Barcelona Supercomputing Center, Barcelona, Spain

**Correspondence to: Cheng You (cyoupuguan@gmail.com)**

**Abstract:**

Arctic cyclones play a crucial role in shaping Arctic weather patterns and influencing sea ice concentrations. Notably, lee cyclogenesis—typically associated with large topographic barriers—has not been observed on the lee side of Greenland, despite its dominance as the Arctic's largest terrain feature. During the MOSAiC expedition in April 2020, an Arctic cyclone was observed at the leeside of Greenland, prompting our hypothesis that lee cyclogenesis contributed significantly to its development.

To test this hypothesis, we conducted simulations with modified Greenland topography. The results confirm that lee cyclogenesis does occur and significantly enhances cyclone intensity. Notably, even when lee cyclogenesis is absent, the jet streak alone sustains cyclone development, suggesting that in this case, both mechanisms—lee cyclogenesis and the jet streak—collectively drive cyclogenesis.

Further analysis reveals the quasi-barotropic nature of lee cyclogenesis. Once the cyclone moves away from Greenland, lee cyclogenesis weakens markedly in the lower troposphere. However, the upper-tropospheric low vortex—induced by orographic forcing—persists, sustaining the cyclone until its dissipation in the central Arctic four days later. This suggests that orographic forcing has a prolonged impact in the upper troposphere. Our findings provide new insights into the mechanisms governing polar cyclone development.



## 1 Introduction

Arctic cyclones are synoptic-scale cyclones that originate either within the Arctic region or migrate into it (Gray et al., 2021; Sepp and Jaagus, 2011; Zhang et al., 2023). They play a significant role in shaping Arctic weather (Fearon et al., 2021) and the sea ice concentrations (Finocchio et al., 2022; Schreiber and Serreze, 2020; Valkonen et al., 2021). Basically, Arctic cyclones share common dynamical mechanisms with their extratropical counterparts.

Upper tropospheric jet streak is one of the important drivers for extratropical cyclones, particularly at the left-hand side of the jet streak's exit (JAMES and HOLZWORTH, 1954; Pinto et al., 2009; Riehl, 1948; Riehl and Teweles, 1953) and at the right-hand side of the jet streak's entrance (Evans et al., 1994; Sinclair and Revell, 2000). Generally, low-pressure systems are more commonly associated with the left-hand side of the jet streak's exit than with the right-hand side of the jet streak's entrance (Achtor and Horn, 1986; Sinclair and Revell, 2000). Jet streaks also contribute to the development of Arctic cyclones, but their role remains primarily supportive according to previous studies (Ban et al., 2023; Qian et al., 2023; Tao et al., 2017).

Lee cyclogenesis, characterized by cyclonic systems emerging on the lee side of mountains, is another significant driver of extratropical cyclone formation. Although extensively documented in regions like the Alps (Buzzi et al., 2020; Buzzi and Tibaldi, 1978) and Rocky Mountains (Chung et al., 1976; Chung and Reinelt, 1973; McClain, 1960). Lee cyclogenesis has not been observed on the lee side of Greenland, despite its prominence as the largest terrain in the Arctic. During the MOSAiC expedition (Multidisciplinary Drifting Observatory for the Study of Arctic Climate; Shupe et al., 2022), an Arctic cyclone was observed east of Greenland in April 2020, positioned at the left side of a jet streak´s exit. This observation led to our hypothesis that both lee cyclogenesis and the jet streak likely played pivotal roles in the development of this cyclone. To test this hypothesis, we have conducted several numerical simulations. The rest of this paper is organized with model set-up in Section 2, followed by results in Section 3, discussion in Section 4, and summary in section 5.



## 2 Model Set-up

In this study, the ICON model (Zängl et al., 2015) is set up as limited area model over a pan-Arctic domain, covering north of $58°$ $N$, at a horizontal resolution of nearly 10 km (R02B08 in ICON terminology). The initial conditions and lateral boundaries are provided by global ICON forecast runs. For more details about the model setup, refer to Bresson et al. (2022) and Kirbus et al. (2023).

Our modeling approach aims at investigating the influence of Greenland's topography on cyclone formation. Therefore, we perform experiments under two scenarios: one scenario with Greenland's topography as it is, and one with Greenland's topography completely removed. It's important to note that we do not alter the surface land use in Greenland. These scenarios are implemented for two nudged runs: ***nudg_8km*** and ***nudg_4km***, and they are referred to as ***nudge_8km_1*** and ***nudg_8km_0*** for run ***nudg_8km***, and as ***nudg_4km_1*** and ***nudg_4km_0*** for run ***nudg_4km***, respectively. These simulations are summed up in Table S1 in the Supporting Information. Run ***nudg_8km*** aims to extract the effects of Greenland's topography from those of the jet streak, while run ***nudg_4km*** seeks to differentiate between the influences of the upper troposphere (above 4 km) and the lower troposphere (0-4 km).

Runs ***nudg_8km*** and ***nudg_4km*** both start on 12 November 2019 at 18:00 UTC, and extend for a duration of 120 hours, concluding on 17 November 2019 at 18:00 UTC. Nudging was applied to horizontal wind, pressure, and temperature, above 4 km for ***nudg_4km*** and 8km for ***nudg_8km***, to hold the dynamical and thermal conditions according to the forcing from the ICON global run. The nudging coefficient increases with height. The atmospheric conditions will rapidly synchronize with those in the ICON global run. In run ***nudg_8km***, the nudging is implemented to maintain the jet streak in accordance with the ICON global run. Conversely, in run ***nudg_4km***, the nudging is intended to preserve the atmospheric circulations above Greenland's topography which reaches a maximum elevation of nearly 3.7 km.

## 3. Results

### 3.1 Synergy between Lee Cyclogenesis and Jet Streak



In simulation ***nudge_8km_1***, a cyclone emerges along the east coast of Greenland from April
13, 2020 (refer to Figure 1). Notably, this cyclone forms on the lee side of Greenland,
coinciding with the presence of a ridge over the region, indicative of lee cyclogenesis.
Furthermore, the cyclone in ***nudge_8km_1*** is positioned to the left of the intensifying jet
streak's exit (Figure 1 a-d). After the jet streak passes beyond April 14, 12:00, the cyclone no
longer remains at the jet streak's exit, leading to its weakening (Figure 1 f-h). This underscores
the significant role played by the jet streak in cyclone development.

To validate the existence of lee cyclogenesis, we conducted a simulation with Greenland's
topography set to zero (***nudge_8km_0***). In nudge_8km_0, the disappearance of the ridge over
Greenland and the subsequent weakening of the cyclone confirms the presence of lee
cyclogenesis (refer to Figure 2). However, even without lee cyclogenesis, the cyclone still
develops, albeit with reduced intensity, indicating that in ***nudge_8km_0***, the jet streak
predominantly influences cyclone development. Therefore, the sea level pressure in
nudge_8km_0 (referred to as $SP_{8km\_0}$) reflects the impact of the jet streak (Figure 2). Without
lee cyclogenesis, the cyclone dissipates more rapidly in ***nudge_8km_0*** compared to
***nudge_8km_1***.

**3.2 Variation of Lee Cyclogenesis**

Based on the sea level pressures in simulations ***nudge_8km_0*** and ***nudge_8km_1***, the intensity
of overall lee cyclogenesis (referred to as $LC_{total}$) is quantified by the difference between sea
level pressure in ***nudge_8km_1*** (referred to as $SP_{8km\_1}$) and sea level pressure in ***nudge_8km_0***
(referred to as $SP_{8km\_0}$) (referred to as $SP_{8km\_0}$, see Eq. 1). Similarly, the difference in
geopotential height at 500 hPa (GH) between these two simulations quantifies the response of
the upper troposphere to Greenland's orographic forcing (referred to as $GH_{total}$; see Eq. 2).

Initially, $LC_{total}$ is confined to the east coast of Greenland (refer to Figures 3a and 3b) on April
13, 09:00. As the cyclone intensifies, the downslope wind extends from the east coast to the
south coast, and $LC_{total}$ extends beyond the east coast, reaching the lee side of the south coast
of Greenland (refer to Figures 3c to 3e). $LC_{total}$ peaks at the south coast of Greenland on April
14, 12:00. It weakens as the cyclone moves away from Greenland (refer to Figures 3f to 3h).
Afterwards, the maximum center of $LC_{total}$ moves back to the east coast where the cyclone
center is located. Meanwhile, $GH_{total}$ intensifies from April 13, 09:00, to April 14, 12:00, at



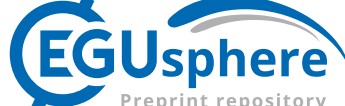

both the east and south coasts, aligning with the variation of $LC_{total}$, indicating the quasi-
barotropic development of lee cyclogenesis.

After April 14, 21:00, although $LC_{total}$ and $GH_{total}$ weaken at the south coast, their persistence
for 4 days without significant dissipation at the east coast suggests a prolonged influence of lee
cyclogenesis in the upper troposphere. We assume that the memory of orographic forcing at
the upper troposphere is longer than that at the lower troposphere. This phenomenon will be
further discussed in section 3.3.

**3.3 Prolonged Influence of Lee Cyclogenesis in the Upper Troposphere**

To test the assumption raised in 3.2, we conducted simulations (***nudge_4km_1*** and
***nudge_4km_0***) to isolate Greenland's orographic forcing at the lower troposphere from its
counterpart at the upper troposphere. The difference in sea level pressure between simulations
***nudge_4km_1*** and ***nudge_4km_0*** represents lee cyclogenesis in the lower troposphere,
denoted as $LC_{low}$ (refer to Eq. 3), while the difference between $LC_{total}$ and $LC_{low}$ quantifies lee
cyclogenesis in the upper troposphere, referred to as $LC_{up}$ (refer to Eq. 4). Similarly, the
difference in geopotential height at 500 hPa between simulations ***nudge_4km_1*** and
***nudge_4km_0*** quantifies the response of the upper troposphere to Greenland's orographic
forcing in run ***nudge_4km*** (referred to as $GH_{low}$; see Eq. 5). The difference between $GH_{total}$
and $GH_{low}$ also represents lee cyclogenesis in the upper troposphere (see Eq. 6). $GH_{low}$ is
almost zero in Figure 4, so $GH_{up}$ is approximately equal to $GH_{total}$. This reveals that nudging
works very well in run ***nudge_4km***. The definitions of $LC_{total}$, $GH_{total}$, $LC_{up}$, $GH_{up}$, $LC_{low}$,
$GH_{low}$ are more clearly explained in following Eq. 1~6,

$LC_{total} = SP_{8km\_1} - SP_{8km\_0}$                                    **(1)**
$GH_{total} = GH_{8km\_1} - GH_{8km\_0}$                              **(2)**
$LC_{low} = SP_{4km\_1} - SP_{4km\_0}$                                   **(3)**
$LC_{up} = LC_{total} - LC_{low}$                                          **(4)**
$GH_{low} = GH_{4km\_1} - GH_{4km\_0}$                            **(5)**
$GH_{up} = GH_{total} - GH_{low}$                                       **(6)**

where $SP$ represents sea-level pressure, while $GH$ denotes the 500 hPa geopotential height and
$LC$ characterizes the intensity of lee cyclogenesis. The subscript "total," "up," and "low" denote



different atmospheric layers, with "total" representing the entire troposphere, "up" referring to
the upper troposphere (4–8 km), and "low" indicating the lower troposphere (0–4 km). The
subscripts "4km_1", "8km_1", "4km_0", and "8km_0" represent simulations ***nudge_4km_1***,
***nudge_8km_1***, ***nudge_4km_0***, ***nudge_8km_0*** respectively**.**

As illustrated in Figure 4, **LC_low** increases from the beginning and peaks on April 14, 12:00,
before gradually diminishing afterwards. As discussed in section 3.2, Lee cyclogenesis
develops quasi-barotropically, inducing a negative **GH_up** at 500 hPa. The variation of **GH_up**
aligns with that of **LC_up** (refer to Figure 5), and they both increase steadily from the beginning
until April 14, 12:00, and maintain their intensity, thereafter, suggesting that the orographic
forcing reduces GH at 500 hPa and forms a low vortex at 500 hPa which afterwards in turn,
maintains a negative SP at the surface.

April 14, 21:00, is a transition time slot. Before April 14, 21:00, the variation of **LC_total** aligns
with that of **LC_low,** while after April 14, 21:00, the variation of **LC_total** aligns with that of **LC_up.**
This suggests that before April 14, 21:00, Lee cyclogenesis is dominated by the orographic
forcing in the lower troposphere, while after April 14, 21:00, the orographic forcing in the
upper troposphere dominates Lee cyclogenesis. This confirms our hypothesis in section 3.2
that the longer memory of Greenland orographic forcing in the upper troposphere, where its
influence persists and dominates the variation of SP until the cyclone dissipates in the central
Arctic on April 18, 2020.

**4 Summary and discussion**

The study investigates the interaction between Lee cyclogenesis and the Jet Streak over
Greenland using numerical simulations. This mechanism is summarized in Figure 6. In the first
part, the synergistic relationship between Lee cyclogenesis and the Jet Streak is examined.
Results show that the presence of Lee cyclogenesis, coupled with the Jet Streak, significantly
influences cyclone development along the east coast of Greenland (Figure 6 a and b). The
weakening of the cyclone after the passage of the Jet Streak suggests its crucial role in cyclone
formation (Figure 6c). Further analysis confirms the existence of Lee cyclogenesis in run
***nudg_8km*** with varying Greenland topography. Even in the absence of Lee cyclogenesis,
cyclone development occurs, albeit with reduced intensity, indicating the dominant influence



of the Jet Streak in cyclone formation. This underscores the complex interplay between
atmospheric dynamics and topographical features.

By isolating Greenland's orographic forcing in run ***nudg_4km***, the study confirms that Lee
cyclogenesis contributes to a low-pressure system both in the lower troposphere (Figure 6a)
and in the upper troposphere (Figure 6b). After the cyclone moves away from Greenland, Lee
cyclogenesis at lower troposphere weakens dramatically, while the low vortex induced by the
orographic forcing in the upper troposphere sustain a low-pressure system in the lower
troposphere (Figure 6c) until it dissipates in the central Arctic after 4 days. The cyclone
effectively transported warm air masses into the Arctic, raising surface temperatures from -
30°C to near melting conditions, signaling the onset of spring (Kirbus et al., 2023; Svensson et
al., 2023).

Compared to Lee cyclogenesis cases observed in the Alps (Buzzi et al., 2020; Buzzi and Tibaldi,
1978) and the Rocky Mountains (Buzzi et al., 2020; Buzzi and Tibaldi, 1978), this case stands
out due to its interaction with a jet streak. While both Lee cyclogenesis and the jet streak play
significant roles, the jet streak is particularly crucial during the intensification phase, whereas
the upper-tropospheric memory of Lee cyclogenesis becomes important for propagation after
the cyclone moves away from the east coast of Greenland. Such synergy between Lee
cyclogenesis and a jet streak is rare in other regions. During the MOSAiC expedition, we
observed the same mechanism not only in this case but also in another instance in May at the
same location. However, unlike the present case, the May cyclone moved toward Scandinavia
instead of the central Arctic.
The persistence of a cyclone for four days and its propagation into the central Arctic raise
questions about additional driving factors. We examined whether the strong temperature
gradient at the ice edge contributed to its longevity, but our results suggest minimal impact
when the ice edge was removed in the simulation. Another possible factor is the low energy
dissipation in the Arctic. The small Rossby Radius at high latitudes indicates reduced energy
dissipation, which may have played a role.
This study proposes a novel mechanism for polar cyclone development, though the frequency
of this process remains unclear. A climatological analysis of lee cyclones near Greenland
would offer valuable insights into future research. The simplified separation of upper- and
lower-tropospheric influences elucidates the synergy between Lee Cyclogenesis and Jet Streak,



however, may oversimplify the complexity of these highly nonlinear atmospheric processes
involved.

**Data Availability**

The ICON-GLOBAL and ICON-LAM (input and output) model data are stored at the AWI
computing centre and are available upon request from the corresponding author.
**Author contributions**
CY prepared the paper, figures and table.
**Competing interests**
The author has declared that there are no competing interests.
**Acknowledgements**
This work was funded by the German Federal Ministry of Education and Research via the
project "Synoptic events during MOSAiC and their Forecast Reliability in the Troposphere-
Stratosphere System (SynopSys)" with grant 03F0872A. The author wants to acknowledge
AWI for providing the technical infrastructure to perform the model runs. The author is grateful
for the helpful discussions with Annette Rinke and Ralf Jaiser.

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

Table 1.  modified parameters in numerical experiments.


| Numerical runs | Topography | |
|---|---|---|
| | 1 | 0 |
| *nudg_8km* | *nudg_8km_1* | *nudg_8km_0* |
| *nudg_4km* | *nudg_4km_1* | *nudg_4km_0* |


















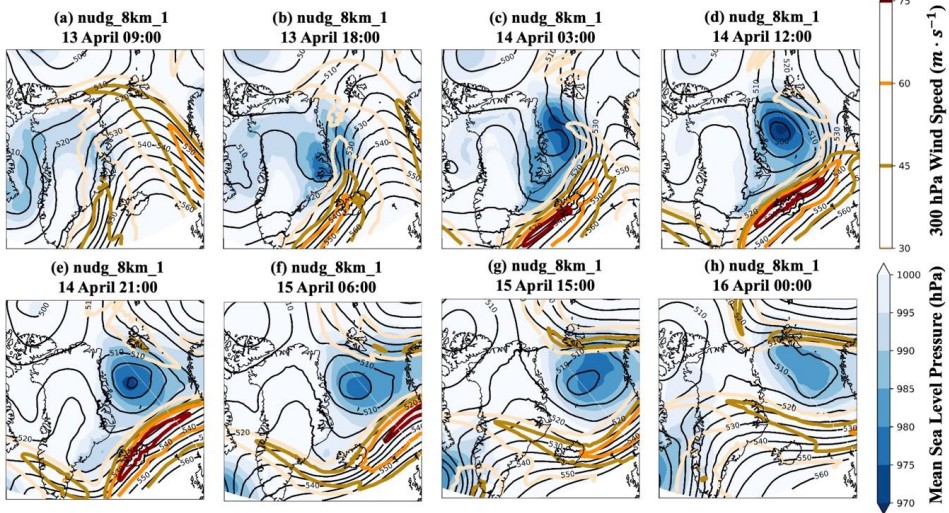


Figure 1: Nine-hourly evolution of surface pressure (color-filled contours; hPa), 500 hPa

geopotential height (black contours; 10 gpm) and 300 hPa wind velocity (colorful contours;

$m \cdot s^{-1}$) in simulation **nudg_8km_1** where nudging is implemented above 8 km to maintain

the jet streak in accordance with the ICON global run.






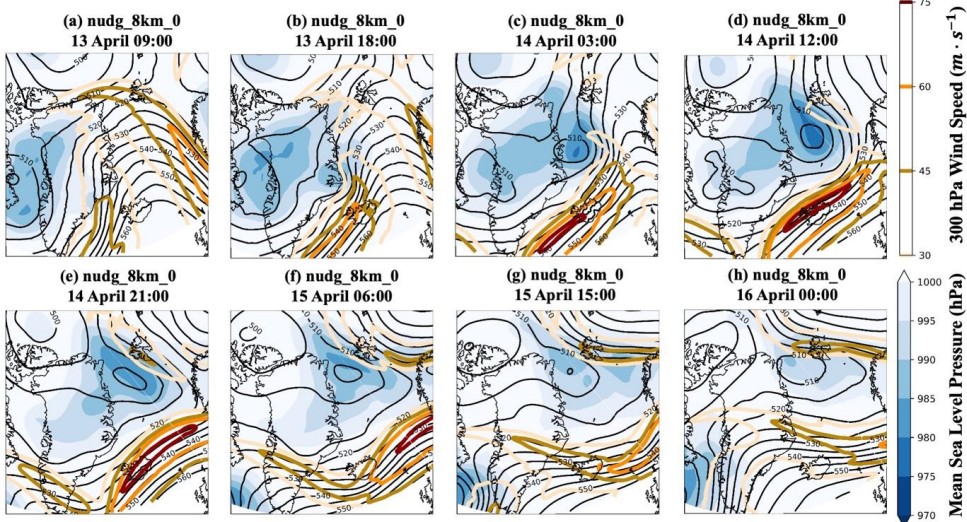


Figure 2: Similar as Figure 1 but for simulation nudg_8km_0 where nudging is implemented

above 8 km and Greenland's topography is set to zero.


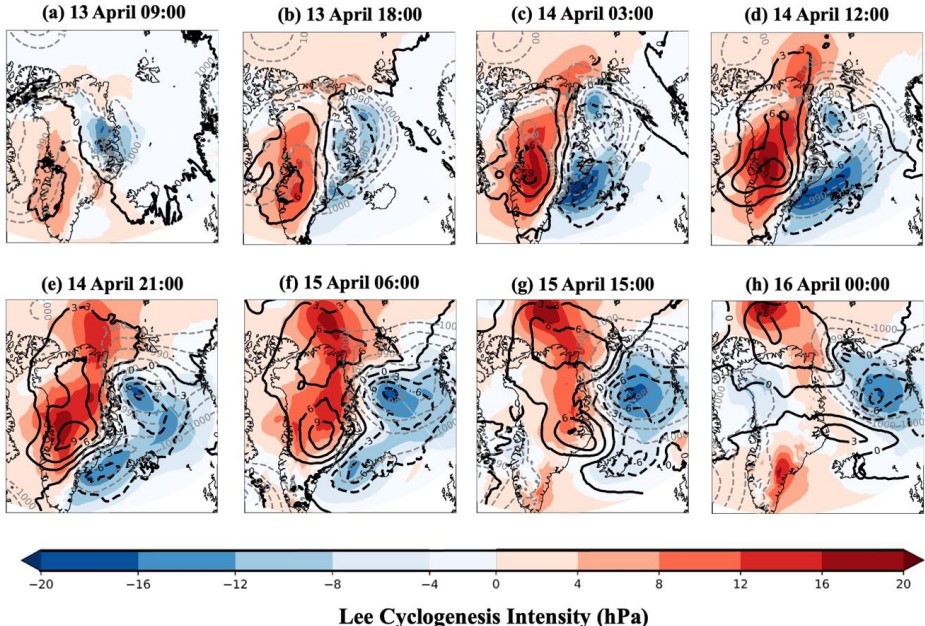



Figure 3: Nine-hourly evolution of the overall lee cyclogenesis $\mathbf{LC_{total}}$ (color-filled contours;

hPa) and Greenland's upper-tropospheric orographic forcing $\mathbf{GH_{total}}$ (black contours; 10gpm).

$\mathbf{LC_{total}}$ is quantified by the sea level pressure difference between simulations ***nudge_8km_1***





and *nudge_8km_0*. Similarly, **GH**$_{total}$ is quantified by 500 hPa geopotential height difference
between simulations *nudge_8km_1* and *nudge_8km_0.*



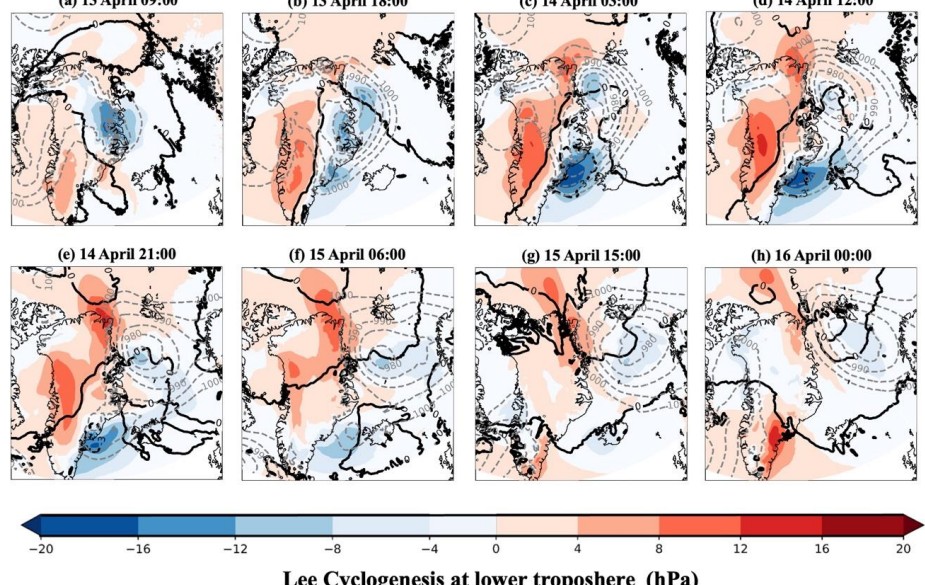



Figure 4: Nine-hourly evolution of the lee cyclogenesis at lower troposphere **LC**$_{low}$ (color-
filled contours; hPa) and Greenland's upper-tropospheric orographic forcing **GH**$_{low}$ (black
contours; 10gpm). **LC**$_{low}$ is quantified by the sea level pressure difference between simulations
*nudge_4km_1* and *nudge_4km_0*. Similarly, **GH**$_{low}$ is quantified by 500 hPa geopotential
height difference between simulations *nudge_4km_1* and *nudge_4km_0.*





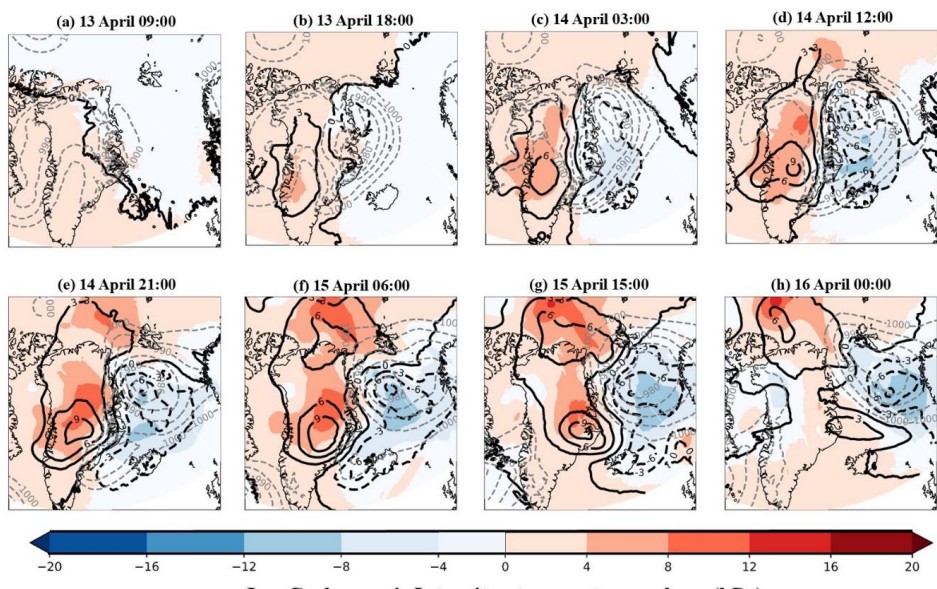

Figure 5: Nine-hourly evolution of the overall lee cyclogenesis $LC_{up}$ (color-filled contours; hPa) and Greenland's upper-tropospheric orographic forcing $GH_{up}$ (black contours; 10gpm). $LC_{up}$ is quantified by $LC_{total}$ - $LC_{low}$. Similarly, $GH_{up}$ is quantified by $GH_{total}$ - $GH_{low}$.





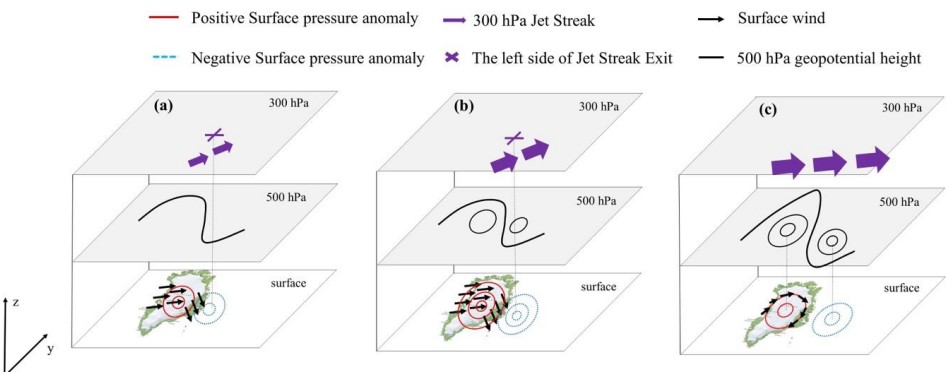



**Figure 6. Schematic diagram of the cyclone development to the east of Greenland.** (a) The
eastern slope of Greenland is located on the left side of the 300 hPa jet stream's exit. Upper-
level divergence, combined with lee cyclogenesis, triggers cyclone development. The Lee
cyclogenesis develops quasi-barotropically, inducing a low-pressure system at both lower and
upper troposphere. These two effects are quantified respectively in Figure 2 and 3; (b) The 300
hPa jet stream intensifies, enhancing its influence on the cyclone. (c) the contribution from the
jet stream disappears since the cyclone is no longer located on the left side of the jet stream
exit, and the lee cyclogenesis weakens from the bottom up as the cyclone moves away from
the eastern coast. At this stage, the upper-troposphere vortex at 500 hPa reaches its peak,
sustaining the surface cyclone for another 4 days and steering it all the way to the central Arctic.





