# Peer review of "Greenland's Topography Triggers Cyclogenesis: Synergy between Lee Cyclogenesis and Jet Streak 3 Cheng You1, 2 4 5 1Alfred Wegener Institute, Helmholtz Centre for Polar and Marine Research, Potsdam, 6 7 8 2 Barcelona Supercomputing Center, Barcelona, Spain 9 10 Co"

_EGUsphere, 2025_

## Referee Comment (RC1)

**"Greenland's Topography Triggers Cyclogenesis: Synergy between Lee Cyclogenesis and Jet Streak"**

**Author:** You

**Recommendation:** Reject

**Overview:**

This study utilizes a series of nudging experiments to examine the role of topography for inducing the development of a lee cyclone during the MOSAiC field campaign. The authors find that the removal of topography strongly limits the development of the surface cyclone, and that the upper-level effects of lee cyclogenesis help to encourage the persistence of the cyclone for several days after the cyclone developed. While there are several interesting aspects to the author's analysis, unfortunately, I find the novelty of the work to be strongly overstated, the methodology is lacking critical details to ensure the reproducibility of the work, the methodology does not consider the sensitivity of results to the adopted modeling approach, and more detailed, quantitative analyses can be performed to better carve out the uniqueness of lee cyclogenesis dynamics near Greenland compared to other regions and to further quantify the influence of orographic processes on the cyclone's development and persistence. Given that these recommendations will likely require substantial changes to the manuscript, I must unfortunately recommend rejection of the manuscript at this time.

**General Comments:**

1. Key details are missing from the methodology section, which unfortunately complicates my ability to evaluate the veracity of subsequent analyses and inhibit the reproducibility of results. These methodological complications are discussed below:

   a. Details regarding the model set-up are rather sparse. For example, the author only provides information regarding the model's initial conditions and the horizontal grid spacing, but no information is provided regarding the number of vertical levels and their spacing, which are presumably very important to a study on orographic effects. Furthermore, there is no discussion of the parameterizations used within the model. In order to make this study fully contained, and given the study relies extensively on model output, more detail about the model set-up should be provided either in the main manuscript or a supplementary section. Additionally, the author is encouraged to provide information about a run script to reproduce the model results, or take efforts to make parts of the model dataset more widely available instead of through a request to the author.

   b. A key aspect of the methodology is the removal of topography, but few details are provided as to how the topography is removed. For instance, removal of the topography requires modification and assignment of atmospheric variables to grid points that lie below ground in the topographic simulations. There are many ways to do this interpolation, and each approach likely will have a substantial influence

on the subsequent results. At the very least, the authors should consider providing information on how the topography was removed, how atmospheric data was interpolated to points that were below ground, and perform sensitivity tests to ensure that their results are robust against whatever interpolation approach is adopted.

c. A key component to a modeling study is to ensure that the resultant circulations bare similarity to a verification dataset. However, only model results from the four experiments are shown. It would be beneficial to include a figure that shows the evolution of the case either within ICON analyses, a separate reanalysis dataset, or within a model simulation nudged everywhere to ICON analyses in order to provide context for how the model simulations deviate from what was actually observed.

d. The authors choose 4 km and 8 km as their cut-off levels for their nudging experiments. 4 km has more justification given that the height of the Greenland Ice Sheet extends to 3.7 km. However, the altitudes of wind speeds associated with a jet streak can extend throughout a substantial vertical depth of the troposphere and well below 8 km. I am curious whether the results are sensitive to this choice of 8 km. Additionally, no details are provided regarding how the nudging exercises were performed other than a few qualitative statements that make it challenging to reproduce results. Last, the results could be very sensitive to the time of initialization, especially since a cyclone is already evident less than 24 h after initialization in the **nudge_8km_1 run**.

2. In its current form, the paper's primary contribution, from my perspective, is to highlight the role of lee cyclogenesis downstream of Greenland. This results, on its own, does not rise to the level of publication for a high-impact journal such as *ACP*. Namely, existing climatologies already highlight areas southeast of Greenland and immediately adjacent to its topography as a local maximum in cyclone frequency (i.e., Fig. 1d from Sprenger et al, 2017; https://doi.org/10.1175/BAMS-D-15-00299.1) and Egger (1974) considered the processes conducive to lee cyclogenesis in Greenland (https://doi.org/10.1175/1520-0493(1974)102<0847:NEOLC>2.0.CO;2). A quick search of AMS journal articles also highlights a long list of articles that discuss and examine lee cyclogenesis in the vicinity of Greenland over the past 40 years, but these studies are not considered extensively in the manuscript. Therefore, I strongly disagree with the assertion that lee cyclogenesis has not been studied before.

3. Following from the previous comment, there are a number of ways the author can leverage their existing analyses and expand them to construct an interesting study that would be suitable for publication. Such an approach might consider calculating a vorticity budget in the vicinity of the cyclone center to identify the role of subsidence and upward vertical motion induced by upper-level divergence associated with the jet streak (i.e., vertical stretching) to more closely isolate the effects of such processes on surface cyclone intensification. They might also consider expanding their analyses to more cases to build out a more robust climatology, construct a larger ensemble of simulations of

varying topographic heights, or perform a series of modeling studies that compare differences in observed cyclone intensification between Greenland, the Rocky Mountains, and the Alps, perhaps. Indeed, one of the unique aspects that sets Greenland apart is the adjacent ocean surface. Investigations could focus on the added role that latent and sensible heat fluxes from different low-level leeside boundaries may play on a lee cyclone's evolution. However, I disagree with the author's comment that a jet streak does not play a role in contributing to lee cyclogenesis in regions outside of Greenland, as prior climatological studies in the Rocky Mountains do show evidence of an attendant jet streak during some of the more impactful events in that region (see comment on L216–222 below). These additional diagnostic approaches and analyses described above are options the author may consider to further bolster their analyses as part of a revision.

**Specific Comments:**

*Abstract*
L16 (and L57): I might recommend softening the language here and elsewhere in the manuscript, as the text suggests that lee cyclogenesis has never occurred here before – when I'm fairly positive a thorough examination of reanalysis data would suggest it is rather ubiquitous under a variety of westerly upper-level flow regimes. Indeed, a quick search of AMS journals highlights numerous articles that consider lee cyclogenesis near Greenland. Instead, I think it might be more accurate to state that the role of lee cyclogenesis has received less consideration in this region compared to other more well studied locations.

*1. Introduction*
L39–41: Consider expanding these statements with a bit more detail, if possible, to further motivate the importance of this work. Namely, what are the effects of Arctic cyclones on sea-ice concentrations? What are their common characteristics relative to midlatitude cyclones, etc.

L49–51: This statement is a bit confusing to me. Namely, what do you mean that the role of the jet streak is largely supportive and how is this different compared to midlatitude cyclones?

L53: More detail could be provided in this paragraph to offer some conceptual foundation as to how lee cyclones develop in a manner that is different from other classes of cyclones. This type of discussion can foreshadow the subsequent diagnostics applied later in the manuscript and those described as part of my third general comment above.

*2. Methods*
L73–75: Given that the model set-up is a crucial component of the forthcoming analyses, I recommend including more detail here instead of referencing prior studies. For example, did these prior studies simulate the same case under consideration here, or did they just adopt a similar model set-up? What were the details of the initial conditions (i.e., resolution)? How many vertical levels were utilized? What are the details concerning model parameterizations? How were the data nudged quantitatively? The answers to these questions can be very relevant for interpreting the forthcoming results and for ensuring the study's reproducibility.

L79: The choice to remove Greenland requires substantial changes to the atmospheric profile at grid points that were below ground level in the topographic simulations. What assumptions were made regarding atmospheric parameters at levels that were initially below ground but no longer are below ground in the no-topography runs?

L81: The naming conventions do not seem rather intuitive at this point in the manuscript and might be best moved until after the next paragraph, which introduces more details concerning the model runs.

L88: When during this time period did the lee cyclone develop? Was there any sensitivity of results to the choice of initialization time? Was there any sensitivity of results to your choice of the number of vertical levels for the nudging experiments? How did the nudging parameter specifically change with altitude?

*3. Results*
L100: Before diving into the simulations, it might be useful to show the actual evolution of the case from a reanalysis product or ICON analyses to demonstrate that the representation of the cyclone within this experimental environment is consistent with what was observed.

L103–104: It is unclear how this flow pattern is indicative of lee cyclogenesis, as the cyclone is upstream of the trough axis and beneath the left jet-exit region, which are favorable synoptic-scale conditions for lee cyclogenesis.

L111–113: It still appears at the start time of the evolution for **nudge_8km_0** that a ridge is located just east of Greenland, just like it is in **nudge_8km_1**. Could you clarify more what is meant by the disappearance of the ridge and how this relates to lee cyclogenesis?

L125: This equation might be more effective if it were introduced coincident with the text described in this paragraph rather than later into section 3.3.

L134: Consider adding a figure panel reference to this statement to help direct the reader's attention accordingly.

L141–142: Consider expanding more on this assumption, in terms of the physical processes that allow you to make this assumption.

*4. Summary and Discussion*
L216–222: I strongly disagree with this statement, as prior studies suggest that strong lee cyclogenesis events downstream of the Rocky Mountains are also accompanied by a jet streak that compliments the cyclone's development (e.g., Winters and Walker 2022; their Fig. 10).

Winters, A. C. and C. L. Walker, 2022: A jet-centered framework for investigating High Plains winter storm severity. *J. Appl. Meteor. Climatology*, **61**, 709–728, doi: 10.1175/JAMC-D-21-0211.1

*Figures and Tables:*

Table 1: Consider adding more information about the details of each model run as part of the table (i.e, parameterizations, resolution, nudging, vertical levels, etc.)

Fig. 3 and subsequent figures: Consider indicating in the caption what the gray dashed contours correspond to.

---

## Referee Comment (RC3)

**Review of "Greenlands Topography Triggers Cyclogenesis: Synergy between Lee Cyclogenesis and Jet Streak" by Cheng You submitted to ACP**

**General comments:**

In this paper a study is presented in which the respective roles of Greenland's topography and an upper-level jet streak in a case of lee cyclogenesis are determined through model experiments in which the topography is removed while parts of the atmosphere are nudged to maintain the original flow. The experiments themselves are interesting in terms of design and some potentially interesting results are presented. However, the novelty of the research is very overstated (and associated literature not cited), methodological details are insufficiently explained, and the analysis lacks depth. The presentation of the manuscript also needs improvement. Hence, I recommend that this manuscript be rejected.

**Major specific comments:**

**Abstract and L56** Here it states "Notably, lee cyclogenesistypically associated with large topographic barriershas not been observed on the lee side of Greenland". This statement is false. For example see Mc Innes et al. (2009, https://doi.org/10.1002/qj.524) which presents an analysis of the mesoscale structure of a mature lee cyclone southeast of Greenland that was observed during a flight with a research aircraft during the Greenland Flow Distortion experiment(GFDex). There are also several other studies that have documented lee cyclogenesis due to Greenland without considering additional local observations from field campaigns (see the introduction to the Mc Innes et al. paper). These studies include studies in which the impact of modifying the orography of Greenland on the cyclogenesis has been assessed (Petersen et al. 2003, https://doi.org/10.1175/1520-0469(2003)060<2183:FITLOI>2.0.CO;2, and Kristjnsson and Mcinnes 1999, https://doi.org/10.1002/qj.49712556003), the same experiment as has been performed in this submitted paper although without the nudging to a global run. Also, many climatologies of extratropical cyclones have shown a pronounced genesis region in the lee of Greenland (e.g., Hodges et al. 2002, 10.1175/1520-0469(2002)059<1041:NPOTNH>2.0.CO;2) whereas in L244 it is implied that such climatologies do not exist: "climatological analysis of lee cyclones near Greenland would offer valuable insights into future research". Hence this work is not as novel as claimed in the abstract and previous literature on lee cyclogenesis due to Greenland needs to be included.

**L40** The statement that "Basically, Arctic cyclones share common dynamical mechanisms with their extratropical counterparts." airbrushes over some important differences between Arctic cyclones and midlatitude cyclones. For example, see the composite analysis of Arctic cyclones by Vessey et al. (2033, https://doi.org/10.5194/wcd-3-1097-2022).

**Methods** When the topography of Greenland is removed how do the fields below the topography height get initialised? The information provided about the model simulations is far too limited. Line 83 refers to a table S1 in the "supporting information". I could not find the supporting information on the ACP webpage (apologies if I have missed this) but there is a Table 1 in the main paper. Is this the table you meant to refer to?

**L130** Please add some additional explanation of the negative pressure difference region that extends from the east coast near the south tip of Greenland towards Iceland. The text refers to downslope winds, which might well exist (though are not explicitly shown), but the relevance of these winds to the lower mean-sea-level pressure in the run with (compared to without) the tropography is not clear.

**Definition of GH$_{total}$ and GH$_{low}$** In the caption for Fig. 3 this it says "Greenlands upper-tropospheric orographic forcing GH$_{total}$" but in the caption for Fig. 4 it says "Greenlands upper-tropospheric

orographic forcing $GH_{low}$". The terms are calculated from different simulations. However using the same description for the two terms is confusing and makes it difficult to interpret what they mean.

**Verification** At no point in this manuscript is the simulated cyclone (for the simulations without the removal of Greenland) compared with reality. The abstract (and later text) make the point that this cyclone "was observed" during the MOSAiC field campaign, so presumably there are local observations that could be used. At the very least though the simulated cyclone should be compared to an operational analysis or to a reanalysis.

**Approach** The approach used in the study, with fields being nudged to "reality" above different heights to infer the impact of Greenland's orographic forcing in both the lower and upper troposphere, is interesting. However, more analysis needs to be presented to demonstrate that the conclusions are valid. In particular, it is not obvious to me that the impact of orographic forcing in the upper troposphere can be inferred from the difference between experiments that purport to demonstrate the impact of orographic forcing on the lower troposphere and whole troposphere given the baroclinic feedbacks between upper and lower levels. An alternative approach could be to consider the quasi-geostrophic forcing from the different levels, for example as in Deveson et al. 2002, https://doi.org/10.1256/00359000260498806.

**Minor specific comments:**

**L50** What does "primarily supportive" mean? As opposed to what?

**L72** Is the resolution nearly 10 km or the grid spacing (as these are different things)?

**Fig. 1** The color-filled contours are presumably mean-sea-level pressure rather than surface pressure (which would be $\sim$700 hPa over Greenland's plateau). Similarly for Fig. 6.

**L123** $LC_{total}$ is defined here but then given in an equation on L161. It would be sensible to move the equations earlier to where the associated terminology begins to be defined.

**L125** $SP_{8km_0}$ is defined twice on this line in addition to on line 116. Also eq. 1 is referred to here well ahead of where the equation is given in the text. Ideally the equation should be on the same page (or an earlier page) to where it is first referred to in the text.

**Fig. 3 and 4** What are the dashed grey contours?

**Fig. 4** The colourbar is labelled "Lee cyclogenesis at lower troposhere (hPa)". This should read "Lee cyclogenesis in the lower troposphere (hPa)" (note spelling error).

**Paragraph beginning L183** It would be helpful if the reader could be pointed to which figure(s) illustrate the points being made in this paragraph. I think we're being asked to compare Figs. 4, 5 and 6.

**L229** Here additional simulations with the ice edge removed are referred to. More details need to be given for the conclusions from these simulations to be included in the paper. Similarly a citation should be added to support that the small Rossby radius at high latitude indicates reduced energy dissipation, possibly Woollings et al. (2023, https://doi.org/10.5194/wcd-4-61-2023).

**Technical errors:**

The English language level in the text has several glitches. I have included a few that I spotted here, though this is not a complete list.

**L43** Grammar. "Upper tropospheric jet streaks are...".

**L43** Why are the author names capitalised in this reference?

**L83** By "summed up" do you mean "summarised"?

**L188** "...forcing in the upper..." should be "...forcing is in the upper..."

**L210** "sustain" should be "sustains".